# Criegee + HONO reaction: the dominant sink of Criegee, and the missing non-photolytic source of OH•

Pradeep Kumar<sup>1</sup>, Vishva Jeet Anand<sup>1</sup>, and Philips Kumar Rai<sup>1</sup>

<sup>1</sup>Department of Chemistry, Malaviya National Institute of Technology Jaipur, Jaipur, 302017, India

**Correspondence:** Pradeep Kumar (pradeep.chy@mnit.ac.in)

- **Abstract.** One of the most important puzzles in atmospheric chemistry is a mismatch between observed and modelled con-
- centrations of OH<sup>•</sup>/HO<sup>•</sup> in the presence of high concentration of volatile organic compounds. It is now well established that
- to fulfill this gap, one needs a reaction that is not only capable of producing OH<sup>•</sup> but also able to act as a sink of HO<sup>•</sup><sub>2</sub>. In the
- present work, we are proposing the Criegee + HONO reaction as a possible solution of this puzzle. Our quantum chemical and
- kinetic calculations clearly suggest that this reaction can not only be an important source of OH radical but can also act as a
- sink of HO<sub>2</sub> radical. Our study also suggests that HONO has the potential to become the most dominant sink of certain Criegee
- intermediates, surpassing SO<sub>2</sub> and water dimer, even in high humid conditions.

## 8 1 Introduction

It is well-known that the atmospheric chemistry is mainly dominated by the radicals (Anderson, 1987; Monks, 2005). Particularly in the troposphere, these radicals are key in degrading various pollutants, a phenomenon as important as the ozone layer 10 for the existence of life (Weinstock, 1969; Lelieveld et al., 2004). The primary radicals responsible for the oxidative power of 11 troposphere come from the  $HO_X$  (OH $^{\bullet}$ , HO $^{\bullet}$ , RO $^{\bullet}$ , RO $^{\bullet}$  etc.) family (Prinn, 2003; Ehhalt, 1987; Khan et al., 2018). Among 12 them, OH• is considered as the most important oxidant in the troposphere (Lelieveld et al., 2002, 2016). Although OH• is the 14 most studied radical in the atmosphere, there are still open questions regarding its sources in the atmosphere (Heald and Kroll, 15 2021; Yang et al., 2024). For a long time, it was believed that OH radicals are mainly formed in daytime via photolysis of 16 tropospheric ozone (O<sub>3</sub>), and nitrous acid (HONO) (Calvert et al., 1994; Alicke et al., 2003; Griffith et al., 2016; Aumont et al., 2003). But now, with various on-field measurements (Geyer et al., 2003; Ren et al., 2003; Emmerson and Carslaw, 2009), it 17 is well established that OH radicals are also present at night in sufficient amounts. In fact, average nighttime concentration of 18  $OH^{\bullet}$  ( $\sim 2.6 \times 10^5$  molecule cm<sup>-3</sup>) is only one order of magnitude lower than its average daytime concentration ( $\sim 1.9 \times 10^6$ molecule cm $^{-3}$ ) (Emmerson and Carslaw, 2009). As the lifetime of OH $^{\bullet}$  is only  $\sim 1$  second, this much concentration of 20 OH• during night indicates its in situ generation via non-photolytic sources. The major non-photolytic source of OH• is the 21 recycling of HO<sup>o</sup> radicals (Whalley et al., 2011; Stone et al., 2012; Hofzumahaus et al., 2009; Smith et al., 2006; Hens et al., 22 2013). Specifically, during the daytime, the primary reaction contributing to this recycling process is  $NO^{\bullet} + HO_{\bullet}^{\bullet}$ , whereas 23 at night, the key reaction is NO<sub>3</sub> + HO<sub>2</sub> (Hall et al., 1988; Mellouki et al., 1988, 1993; Rai and Kumar, 2024). However, 24 compared to photolytic sources, non-photolytic sources of OH• remain less understood in atmospheric chemistry (Brown and 25 Stutz, 2012; Emmerson and Carslaw, 2009). This is evidenced by the fact that, in the atmosphere with a high concentration of volatile organic compounds (VOCs), atmospheric models consistently under-predict the concentration of OH• compared to

the observed value (Emmerson and Carslaw, 2009; Stone et al., 2012). This discrepancy is especially pronounced in winter 28 (Harrison et al., 2006; Heard et al., 2004; Slater et al., 2020) and indoor environments (Østerstrøm et al., 2025; Gomez Alvarez 29 30 et al., 2013; Reidy et al., 2023), where light plays a minimal role. In addition, the discrepancy between measured and observed value of OH• was also found to depend upon NO<sub>X</sub> concentration. Both under low NO<sub>X</sub> (Carslaw et al., 2001; Tan et al., 2001; 31 Lelieveld et al., 2008; Tan et al., 2017) as well as high  $NO_X$  (above 6 ppbv) (Slater et al., 2020), the discrepancy was found to 32 be quite significant. As the primary recycling of  $HO_2^{\bullet}$  to  $OH^{\bullet}$  occurs via  $NO_X$ , the under-prediction of  $OH^{\bullet}$  by models under 33 low NO<sub>X</sub> conditions suggests either the presence of another route for recycling or some new non-photolytic source of OH $^{\bullet}$ . 35 This hypothesis is further strengthened by a few combined experimental and modelling studies. For example, Lu et al.(Lu et al., 2012) have to introduce an artificial source of  $OH^{\bullet} \leftrightarrow HO_{\bullet}^{\bullet}$  inter-conversion ( $RO_{\bullet}^{\bullet} + X \longrightarrow HO_{\bullet}^{\bullet}, HO_{\bullet}^{\bullet} + X \longrightarrow OH^{\bullet}$ ) in 36 their atmospheric model to match the experimental concentration profile. In an another study, to match the experimental OH concentration with models, Whalley et al. (Whalley et al., 2011) increased the concentration of VOCs in their model. Although 38 their computed OH<sup>•</sup> concentration becomes closer to experimental value, the mismatch between observed and measured con-39 40 centration of HO<sub>2</sub> becomes worse. There have been various attempts to identify the missing source of OH<sup>2</sup> in the atmosphere (Paulot et al., 2009; Peeters et al., 2014; Sander et al., 2019). For example, Peeters et al. (Peeters et al., 2009; Peeters and 41 Mu'ller, 2010; Peeters et al., 2014) suggested that the oxidation of isoprene can regenerate  $HO_X$  radicals in the presence of 42 light via isoprene-peroxy radical interconversion and isomerisation pathways (Leuven Isoprene Mechanism (LIM)). Although 43 the introduction of LIM into chemical models were found to improve the value of modelled OH<sup>•</sup> concentration, the modelled 44 45 values still remain under-predicted (Crounse et al., 2011; Teng et al., 2017; Berndt et al., 2019; Novelli et al., 2020; J. Medeiros 46 et al., 2022). Particularly, the LIM is more effective in regions where biogenic volatile organic compounds (BVOCs) dominate and  $NO_X$  concentration is ultra low, e.g. rain forest regions (Whalley et al., 2011; Feiner et al., 2016; Lew et al., 2020). In 47 48 contrast, in regions where sufficient anthropogenic sources of VOCs are present, e.g. in polluted areas, LIM is not effective. In addition, LIM is not fundamentally a HO<sup>o</sup><sub>2</sub> to OH<sup>o</sup> interconversion process, rather it is the recycling of VOCs to OH<sup>o</sup>. In a 49 50 recent study, Yang et al. (Yang et al., 2024) suggested that aldehyde could be an additional source of OH. Authors proposed 51 that the autoxidation of carbonyl organic peroxy radicals (R(CO)O<sub>2</sub>) derived from higher aldehydes, can produce OH• through 52 photolysis (RAM mechanism). Though RAM mechanism efficiently predicts  $OH^{\bullet}$  production at low  $NO_X$  concentrations, it still under-predicts the same at high  $NO_X$  concentrations. Interestingly, when both LIM and RAM are incorporated into a base 53 model in the presence of moderate concentration of  $NO_X$ ,  $OH^{\bullet}$  concentration improves significantly, but the discrepancy in the 54 modelled and observed HO<sup>o</sup> remains unresolved. It is also worth mentioning that photolysis is an important part of both, LIM 55 and RAM, and hence, both of these mechanism do not offer any help in improving the model OH• concentration in nocturnal 56 57 environment. Furthermore, both LIM and RAM are also not directly involved in recycling of HO<sup>o</sup><sub>2</sub> to OH<sup>o</sup>. The discrepancy in the model occurs during both day and night (Faloona et al., 2001; Hens et al., 2013; Geyer et al., 2003), and is associated with 58 59 HO<sup>♠</sup> to OH conversion (Whalley et al., 2011; Hofzumahaus et al., 2009). In light of these studies, we believe that the puzzle of missing  $OH^{\bullet}$  source is very much alive and the key to this puzzle may be a non-photolytic source capable of  $HO_2^{\bullet} \leftrightarrow OH^{\bullet}$ 60 recycling. 61

2

In the present work, we are proposing reaction of Criegee intermediate with HONO as a source of OH<sup>•</sup>. Criegee Intermediates

(CIs) are formed during the ozonolysis of alkenes (Criegee, 1975; Johnson and Marston, 2008; Taatjes, 2017). In fact, alkene 63 ozonolysis is a highly exothermic reaction produces energized CIs. Some of the energized CIs readily convert into OH• via 64 65 unimolecular decomposition, while the remaining CIs get collisionally stabilized (sCI) (Horie and Moortgat, 1991; Donahue et al., 2011; Novelli et al., 2014; Alam et al., 2011), sCIs can undergo either a thermal unimolecular dissociation or a bimolec-66 ular reaction. Depending upon concentration of the co-reactant and rate constant of such bimolecular reaction, the bimolecular 67 68 reaction paths can be the main sink of sCI (Osborn and Taatjes, 2015; Lin et al., 2015; Sheps et al., 2014; Vereecken and Francisco, 2012). There are several studies in the literature that suggest CI reacts rapidly with the trace gases present in the 70 atmosphere (Cox et al., 2020; Mallick and Kumar, 2020; Vereecken et al., 2015). In this work, we are suggesting HONO as 71 a new partner for the bimolecular reaction of Criegee intermediates as a possible source of OH $^{\bullet}$ . The concentration of CI ( $\sim$  $10^4 - 10^5$  molecule cm<sup>-3</sup>) in the atmosphere is comparable with Cl $^{\bullet}$  ( $\sim 5.0 \times 10^4 - 3.0 \times 10^5$  molecule cm $^{-3}$ ) and OH $^{\bullet}$  ( $\sim$  $1.0 \times 10^5 - 4.0 \times 10^6$  molecule cm<sup>-3</sup>) (Khan et al., 2018; Novelli et al., 2017). Similarly, nitrous acid (HONO) is also an im-73 portant trace gas present in the nighttime atmosphere in a considerable amount (Li et al., 2021; Song et al., 2023). The average 74 concentration of HONO is  $\sim 8.9 \times 10^{10}$  molecule cm<sup>-3</sup>, which can reach as high as  $\sim 6.9 \times 10^{11}$  molecule cm<sup>-3</sup> during the 75 fog event (Pawar et al., 2024). Although a general wisdom about HONO is, its concentration builds up in nighttime, and in 76 daytime, it decomposes via photolysis to give OH<sup>•</sup>, HONO itself is a highly reactive molecule and can participate in various 77 bimolecular chemical reactions during night (Anglada and Sole, 2017; Lu et al., 2000; Wallington and Japar, 1989). Moreover, 78 in indoor environments, high concentrations of OH<sup>•</sup> have been found to strongly correlate with high concentrations of HONO 79 (Gomez Alvarez et al., 2013). It is important to mention that, the reaction of HONO with the simple Criegee intermediate 80 81 (CH<sub>2</sub>OO) has already been investigated theoretically (Kumar et al., 2022). In that investigation, the major product was predicted to be hydroperoxymethyl nitrite (HPMN). We will show in the present work that the main product of this reaction is 82 83 OH• and this path is the dominant path of the title reaction.

## 84 2 Methodology

85

## 2.1 Electronic structure theory

There are two parts of electronic structure theory; optimization and subsequent single-point energy calculations. The criteria 87 behind choosing a method for optimization is; it should be computationally not very demanding and at the same time, it 88 should accurately predict the geometries and frequencies of the species involved in the reaction. Based on these criteria, all the geometries have been optimized using M06-2X functional in conjuction with aug-cc-pVTZ basis set using Gaussian16 89 software package (Frisch et al., 2016). We have compared the geometrical parameters of the isolated species obtained at M06-90 2X/aug-cc-pVTZ level of theory with the experimental (Johnson III, 2013; Ruscic et al., 2004) values available in the literature 91 in Figure S1 of the ESI. It is evident from Figure S1 that the maximum deviation in bond lengths at M06-2X/aug-cc-pVTZ level 92 of theory from the experiment was only  $\sim 0.04$  Å, whereas the maximum deviation in bond angles from the experiment was 93  $\sim 1^{\circ}$ . It clearly suggests that M06-2X/aug-cc-pVTZ level of theory is providing accurate geometries of the isolated species. In addition, we have also compared the frequencies of the isolated species obtained at M06-2X/aug-cc-pVTZ level of theory

with the experimental values in Table S2 of the ESI. The maximum deviation in frequency from experiment was  $\sim 250 \text{ cm}^{-1}$ . 96 Therefore, we believe that M06-2X/aug-cc-pVTZ level of theory is appropriate for optimization and frequency calculations. 97 This conclusion is also consistent with the previous work (Kumar et al., 2022) where M06-2X/aug-cc-pVTZ level of theory 98 was found to be adequate for the title reaction. For the second part, we carried out single-point energy calculations for the 99 optimized geometries at CCSD(T) level of theory in complete basis set limit (CBS). To estimate energies at CCSD(T)/CBS 100 101 level of theory, first, we calculated the single point energies at CCSD(T)/aug-cc-pVDZ, and CCSD(T)/aug-cc-pVTZ level of theory, and then extrapolated these energies to corresponding CBS limit using the method of Varandas and Pansini (Varandas 102 103 and Pansini, 2014; Pansini et al., 2016) (see ESI for the details).

#### 2.2 Kinetics

110

111

113

114

117

119

120

121

123 124

126

127

Energetics calculations shed light only on enthalpic requirement of the reaction, for a barrierless process, entropy is an equally 106 important factor. Therefore, to account for both, enthalpy and entropy, we have estimated the rate constant for CH<sub>2</sub>OO + HONO reaction within a temperature range of 213–320 K. 107

The mechanism of CH<sub>2</sub>OO + HONO reaction can be represented by following reaction: 108

$$CH_2OO + HONO \xrightarrow{k_f} RC1 \xrightarrow{K_{uni}} PC1 \longrightarrow CH_2O + OH^{\bullet} + NO_2$$
 (R1)

To calculate the overall rate constant of the title reaction, we have used the master equation approach as implemented in the MESMER software package. It is evident that reaction R1 proceeds in three steps. In the first step, the formation of RC occurs via a barrierless association of isolated reactants. MESMER uses the inverse Laplace transform (ILT) method to estimate the 112 energy-dependent rate constant, k(E), for this step. This, in turn, requires fitted Arrhenius parameters as input to MESMER, which are obtained using KTOOLS code as implemented in the MultiWell suite of programs (Barker et al., 2021). KTOOLS uses variational transition state theory (VTST) for the barrierless reaction. The inputs for KTOOLS are potential energy surface 115 (PES) scans along the coordinate describing the dissociation of RC into isolated reactants. Each point on the PES serves as a 116 trial transition state; KTOOLS searches for the transition state for which the reaction flux is minimized. In the next step, RC 118 undergoes unimolecular dissociation to PC via a transition state. MESMER applies Rice-Ramsperger-Kassel-Marcus (RRKM) theory, including tunneling contributions via an unsymmetrical Eckart barrier. In the final step, PC spontaneously dissociates to form isolated products. It is important to mention that we do not find a tight transition state for product formation from PC; therefore, we have treated this step also using ILT. It is worth noting that the reactant complex (RC) and the transition state 122 (TS) exhibit hindered rotational motions, and multiple conformations may exist due to different torsional angles. To account for this, we have used the HinderedRotorOM1D model in MESMER to compute rate constants. Specifically, we performed a one-dimensional potential energy scan of OH torsion along the N-O bond in both RC and TS, covering the full 0° to 360° 125 range. The resulting energy profiles are used to calculate the hindered rotor partition functions. During this scan, we found local minima in both RC and TS, suggesting that our originally optimized structures correspond to the global minimum conformers. The Lennard-Jones (L-J) model is used to calculate the collision frequency between reactants and the bath gas. Air is used as

the bath gas, with L-J parameters  $\sigma = 3.68$  Å and  $\epsilon/kT = 86.2$  K. The L-J parameters for RC are not known; therefore, they are estimated using the L-J parameters of CI and HONO via the combining rule (Schnabel et al., 2007).

$$\sigma_{RC} = \frac{1}{2}(\sigma_{CH_2OO} + \sigma_{HONO})$$

$$(\epsilon/k_B)_{RC} = [(\epsilon/k_B)_{CH_2OO}(\epsilon/k_B)_{HONO}]^{1/2}$$

- The L-J parameters for RC using above equations turn out to be,  $\sigma$ =4.68 and  $\epsilon$ =246.15 K. A single-exponential down model
- is used to describe the collisional energy transfer probability with a maximum energy grain size of 100 cm<sup>-1</sup> and  $\Delta E_{down} =$
- $150 \text{ cm}^{-1}$ .

## 3 Results and discussion

- In the present work, we have investigated the reactions of Criegee intermediates (CIs) with nitrous acid (HONO). It is known
- that the reactivity of CI is greatly influenced by the substitution group present on carbon center of the CI. Therefore, to account
- for it, we have studied two types of CIs; the simplest Criegee intermediate (CH<sub>2</sub>OO) and the dimethyl-substituted Criegee in-
- termediate ( $(CH_3)_2COO$ ). Another motivation for choosing  $(CH_3)_2COO$  comes from the fact that in contrast to simple Criegee
- which is formed only from the ozonolysis of ethene, the dimethyl-substituted Criegee intermediate can be generated from the
- ozonolysis of many highly abundant alkenes, such as terpenes and mycrene, and hence, the concentration of (CH<sub>3</sub>)<sub>2</sub>COO is
- significantly higher in the atmosphere. In this section, we will first discuss the energetics and kinetics of CH<sub>2</sub>OO + HONO
- reaction, followed by  $(CH_3)_2COO + HONO$  reaction.
- The potential energy surface for CH<sub>2</sub>OO + HONO reaction is depicted in Figure 1. It is evident from Figure 1 that reac-
- tion occurs in two steps; in the first step, CH<sub>2</sub>OO interacts with H atom of HONO via hydrogen bonding and forms a stable
- reactant-complex (RC1), which is  $\sim 10.1$  kcal mol<sup>-1</sup> stable than isolated reactants. In the next step, RC undergoes a uni-
- molecular transformation to form final products, i.e., CH<sub>2</sub>O, OH•, and NO<sub>2</sub>. This happens via a transition-state (TS1) that is
- effectively  $\sim 8.0$  kcal mol<sup>-1</sup> below the isolated reactants. It suggests that the formation of OH $^{\bullet}$  via CH<sub>2</sub>OO + HONO reaction
- is a barrierless process. Prior to the formation of the isolated products, a product complex (PC1) is formed which is  $\sim 44.7$
- 150 kcal mol<sup>-1</sup> stable than isolated reactants. The overall reaction was found to be exothermic by  $\sim 17.3$  kcal mol<sup>-1</sup> that lies close
- to the experimental value of  $\sim 16.9$  kcal mol<sup>-1</sup> (Ruscic et al., 2004), again confirming the adequacy of the methodology used.
- The computed bimolecular rate constant values ( $k_{bi}^{CH_2OO}$ ) for CH<sub>2</sub>OO + HONO reaction in the temperature range 213–320
- K are given in Table 1. It is evident from Table 1 that the values of  $k_{bi}^{CH_2OO}$  slightly decrease with temperature increasing,
- a typical character of a barrierless process. For example, at 213 K, values of  $k_{bi}^{CH_2OO}$  is  $1.17 \times 10^{-11}$  cm<sup>3</sup> molecule<sup>-1</sup> sec<sup>-1</sup>
- which becomes  $\sim 6.3 \times 10^{-12}$  cm<sup>3</sup> molecule<sup>-1</sup> sec<sup>-1</sup> at 320 K.
- Figure 2 depicts the potential energy surface of (CH<sub>3</sub>)<sub>2</sub>COO + HONO reaction. It is evident from Figure 2 that (CH<sub>3</sub>)<sub>2</sub>COO
- + HONO reaction also proceeds in two steps; in the first step, (CH<sub>3</sub>)<sub>2</sub>COO associates with HONO to form a stable reactant-
- complex (RC2) that is  $\sim 14.2$  kcal mol<sup>-1</sup> more stable than isolated reactants. Finally, RC transforms into isolated products,
- i.e.,  $(CH_3)_2CO$ ,  $OH^{\bullet}$ , and  $NO_2$ . This transformation occurs through a transition state that lies  $\sim 10.1$  kcal mol<sup>-1</sup> below the

with stabilization energy  $\sim$  -36.2 kcal mol<sup>-1</sup> with respect to isolated reactant. 161 Using the energetics, we have also computed the rate constant for (CH<sub>3</sub>)<sub>2</sub>COO + HONO reaction employing master equation 162 in the same 213–320 K temperature range. The calculated bimolecular rate constants  $(k_{bi}^{(CH_3)_2COO})$  are listed in Table 1. It 163 is evident from Table 1 that similar to  $\mathrm{CH_2OO}$  + HONO reaction, here also the values of  $\mathrm{k}_{bi}^{(CH_3)_2COO}$  slightly decrease with 164 increasing temperature within whole range of temperature. But the bimolecular rate constant of (CH<sub>3</sub>)<sub>2</sub>COO + HONO reaction 165 becomes  $\sim 2.6$  to 3.6 times higher compared to the same for CH<sub>2</sub>COO + HONO reaction at all temperatures considered in the 166 present work. For example, at 298 K, the value of  $k_{bi}^{(CH_3)_2COO}$  is  $\sim 2.03 \times 10^{-11}$  cm<sup>3</sup> molecule<sup>-1</sup> sec<sup>-1</sup>, whereas the value of 167  $k_{bi}^{CH_2OO}$  is only  $\sim 7.2 \times 10^{-12}$  cm<sup>3</sup> molecule<sup>-1</sup> sec<sup>-1</sup>. In the bimolecular rate constant, the capture rates of both the reactions 168 are almost same, given in Table S3 of the ESI. The difference in the rate values of the two reactions depends on whether the 169 reactant complex will proceed forward or backward, which further depends on the forward and backward Gibbs free energy 170 barriers of the reactant complex. The Gibbs free energy profile at 298 K is shown in Figure S2 of the ESI. It is evident from 171 Figure S2 that due to the higher stabilization of RC2, its reverse free energy barrier is high ( $\sim 2.9 \text{ kcal mol}^{-1}$ ), while the same 172 is very low for RC1 ( $\sim$  -1.3 kcal mol<sup>-1</sup>). Consequently, the relative yields of product are higher for the (CH<sub>3</sub>)<sub>2</sub>COO + HONO 173 reaction compared to CH<sub>2</sub>COO + HONO reaction. 174 Lastly, it is important to discuss the uncertainties associated with the computed rate constant due to limitations in the method-175 ology used for the energetics calculations. For example, a major source of uncertainty can originate from the fact that Criegee 176 177 intermediates are known to possess moderate multireference character, and CCSD(T)/CBS sometimes fails in accurately pre-178 dicting the energetics of such reactions (Rai and Kumar, 2022; Mallick et al., 2019; Mallick and Kumar, 2018). It is worth mentioning that for multireference systems, incorporating higher-level excitations at the coupled-cluster level yield energet-179 180 ics within chemical accuracy (Tajti et al., 2004; Misiewicz et al., 2018; Nguyen et al., 2013; Anand and Kumar, 2023; Rai and Kumar, 2023). To assess the uncertainty in the energetics arising from the multireference character, we have carried out 181 182 CCSDT(Q)/CBS calculations for the smaller Criegee intermediate reaction, i.e., CH<sub>2</sub>OO + HONO. We focused on key station-183 ary points; the reactant complex (RC) and the transition state (TS). The various components of the post-CCSD(T) corrections  $(\delta_T \text{ and } \delta_{T(Q)})$  are provided in Table S7 of the ESI. It is evident from Table S7 that post-CCSD(T) corrections lead to only mi-184 nor changes in the calculated energetics of CH<sub>2</sub>OO + HONO reaction. Quantitatively, these corrections reduce the stabilization 185 energy of RC by  $\sim 0.54$  kcal mol<sup>-1</sup>, while increasing the barrier height by a similar  $\sim 0.67$  kcal mol<sup>-1</sup>. Both variations fall 186 well within the range of chemical accuracy. This supports the reliability and computational efficiency of our chosen level of 187 theory, CCSD(T)/CBS//M06-2X/aug-cc-pVTZ, for studying the title reaction. Another source of uncertainty in the computed 188 rate constant may arise from the error in estimation of frequency. The maximum deviation in frequency is 250 cm<sup>-1</sup>, which 189 corresponds to  $\sim 0.7$  kcal mol<sup>-1</sup> in the reaction energetics. To account for this, we have assumed an uncertainty of  $\pm 1$  kcal 190 191 mol<sup>-1</sup> in both well depths and reaction barriers. Using this assumption, we estimated the resulting uncertainty in the rate constants at 298 K for the model reaction CH<sub>2</sub>OO + HONO. Due to  $\pm$  1 kcal mol<sup>-1</sup> uncertainty in the reaction barriers and 192 well depths, the deviation in the rate constant is  $\sim 7.21^{+4.67}_{-3.65} \times 10^{-12} \text{ cm}^3 \text{ molecule}^{-1} \text{ sec}^{-1} \ (\pm 1 \text{ reaction barriers})$  and  $\sim$ 193  $7.21^{+0.45}_{-0.45} \times 10^{-12} \text{ cm}^3 \text{ molecule}^{-1} \text{ sec}^{-1} (\pm 1 \text{ well depths}), \text{ respectively.}$ 194

isolated reactants, making the overall reaction barrierless. Here also product complex (PC2) is formed before isolated product

160

## 195 4 Atmospheric implications

After estimating the energetics and kinetics of title reaction, it is important to discuss the impact of title reaction in the atmospheric chemistry. The importance of title reaction in the atmosphere critically depends on how it competes with other known 197 sinks of Criegee intermediate, i.e., H<sub>2</sub>O, (H<sub>2</sub>O)<sub>2</sub>, NO<sub>2</sub>, NO, CO, and SO<sub>2</sub>. The efficiency of a chemical reaction in the atmo-198 sphere depends upon two factors; rate of reaction and concentration of co-reactants. The effective rate constant  $(k_{eff})$  captures 199 200 both of these factors as it is defined as the multiplication of bimolecular rate and concentration of co-reactants. Therefore, 201 we have used k<sub>eff</sub> to compare the effectiveness of title reaction compared to other sinks of Criegee intermediates. A list of effective rates for the reaction of CI with H2O, (H2O)2, NO2, NO, CO, and SO2 at 298 K are provided in Table S4 of the ESI. 202 203 To compute  $k_{eff}$ , the average concentrations of all the sinks have been taken from polluted urban environments. The corresponding rate coefficients of all the sinks are taken from experimental measurements. One can see from Table S4, the effective 204 205 rate coefficients  $(k_{eff})$  of CO, NO, and NO<sub>2</sub> are lower compared to those of SO<sub>2</sub>, H<sub>2</sub>O, and  $(H_2O)_2$ . For example,  $k_{eff}$  for the reaction of CI with SO<sub>2</sub> is 3.35 sec<sup>-1</sup>, while that for NO<sub>2</sub> is only 0.9 sec<sup>-1</sup>. Therefore, in the present work, we have focused 206 207 our attention on a detailed comparison of the title reaction with SO<sub>2</sub>, H<sub>2</sub>O, and (H<sub>2</sub>O)<sub>2</sub>. The stabilized Criegee such as unsubstituted and disubstituted Criegee intermediates can dissociate via bimolecular reactions with radicals, depending upon the 208 209 concentration of their co-reactant in the atmosphere. As far as abundance of HONO is concerned, it is found in both regions; forested as well as polluted in significant amounts (Kim et al., 2015; Acker et al., 2006; Ren et al., 2010; Zhang et al., 2012; 210 211 He et al., 2006; Su et al., 2008; Ren et al., 2006; Rondon and Sanhueza, 1989; Zhou et al., 2011; Pawar et al., 2024; Vereecken 212 et al., 2012). Among the two, HONO concentrations are comparatively higher in polluted urban areas, such as megacities. Therefore, we expect HONO to play a more effective role as a sink for Criegee intermediates in such regions. Hence, we have 213 used representative concentrations of HONO and SO<sub>2</sub> in urban areas for the primary comparison. The concentration of water 214 varies greatly in the atmosphere depending upon saturation vapour pressure and relative humidity (RH) (Anglada et al., 2013; 215 Rai and Kumar, 2025). Therefore, in the case of  $H_2O$  and  $(H_2O)_2$ , we have taken two concentrations; one calculated at 20% 216 RH, and the other calculated at 100% RH. The former serves as lower limits of H<sub>2</sub>O and (H<sub>2</sub>O)<sub>2</sub> concentrations, whereas the 217 latter serves as the upper limits of  $H_2O$  and  $(H_2O)_2$  concentrations. 218 In Figure 3, we have compared the  $k_{eff}$  of  $CH_2OO + HONO$  with the  $k_{eff}$  of  $CH_2OO + H_2O/(H_2O)_2/SO_2$  reactions. Figure 3 219 220 shows, at 100% RH,  $k_{eff}$  of CH<sub>2</sub>OO + (H<sub>2</sub>O)<sub>2</sub> is the dominant reaction across the entire temperature range (213–320 K) (Lin et al., 2016). At 20% RH,  $k_{eff}$  for CH<sub>2</sub>OO + (H<sub>2</sub>O)<sub>2</sub> and CH<sub>2</sub>OO + H<sub>2</sub>O remain dominant at higher temperatures, specifically 221 222 within 235–320 K and 260–320 K, respectively. However, at lower temperatures,  $k_{eff}$  of  $CH_2OO + HONO$  becomes domi-223 nant, surpassing both,  $CH_2OO + (H_2O)_2$  and  $CH_2OO + H_2O$  in the range of 213–235 K and 213–260 K, respectively. As far as  $CH_2OO + SO_2$  reaction is concerned (Onel et al., 2021), its  $k_{eff}$  values are  $\sim 5$  times higher than that of  $CH_2OO + HONO$ 224 reaction within the whole temperature range, indicating that CH<sub>2</sub>OO + HONO reaction is a minor contributor compared to 225  $CH_2OO + SO_2$ . 226 Similarly, we have compared our dimethyl substituted Criegee reaction ((CH<sub>3</sub>)<sub>2</sub>COO + HONO) with other known bimolec-227 228 ular reactions of  $(CH_3)_2COO$ . Here also, we have computed  $k_{eff}$  for the comparison (see Figure 4). The rate constants of

 $(CH_3)_2COO + SO_2$  reaction (Smith et al., 2016) is known in the range of 283–303 K, and hence, we have compared its  $k_{eff}$ 229 in this temperature range with dimethyl substituted (CH<sub>3</sub>)<sub>2</sub>COO + HONO reaction. Figure 4 shows that unlike CH<sub>2</sub>OO + 230 231 HONO reaction, here  $k_{eff}$  of  $(CH_3)_2COO + HONO$  is 2 times higher than the same for  $(CH_3)_2COO + SO_2$  reaction within 232 283–303 K. In addition, it is worth mentioning that under certain atmospheric conditions, concentration of HONO can be quite high compared to SO<sub>2</sub>. For example, during fog events, it is well known that concentration of SO<sub>2</sub> drops significantly (Zhang 233 234 et al., 2013) while concentration of HONO increases (Pawar et al., 2024), making HONO a potentially major sink of Criegee intermediates in fog-like environments. In addition, as SO<sub>2</sub> mainly comes from human activities, its concentrations are high 235 236 in polluted areas and become quite very low in tropical forests and rural areas. In fact, its concentrations fall below detection limits in tropical forest regions (Vereecken et al., 2012). In contrast, although HONO concentration is also high in polluted 237 238 regions compared to a clean environment, due to the various in situ sources, HONO is present in reasonable amounts even in tropical forest areas (Zhang et al., 2012). Therefore, in this region also, HONO is a more effective sink of CI compared to SO<sub>2</sub>. 239 240 Moreover, CI + HONO reaction is a hydrogen atom transfer (HAT) process, and hence, the presence of water can effectively 241 catalyze this reaction (Buszek et al., 2012; Viegas and Varandas, 2012; Rai and Kumar, 2025). In contrast, the presence of water, 242 particularly droplets and aerosols, can act as a sink for SO<sub>2</sub> (Zhang et al., 2013), and hence, in the presence of water, Criegee + SO<sub>2</sub> reaction should be less important compared to CI + HONO reaction. After establishing that compared to SO<sub>2</sub>, HONO 243 is a more effective sink for (CH<sub>3</sub>)<sub>2</sub>COO under most of the conditions, at last, it is important to compare it with (CH<sub>3</sub>)<sub>2</sub>COO 244 + H<sub>2</sub>O/(H<sub>2</sub>O)<sub>2</sub> reactions (Vereecken et al., 2017). It is evident from Figure 4 that at 100% RH, k<sub>eff</sub> of (CH<sub>3</sub>)<sub>2</sub>COO + HONO 245 can dominate over  $k_{eff}$  of  $(CH_3)_2COO + H_2O$  and  $(CH_3)_2COO + (H_2O)_2$  for a relatively wider range of temperatures. For 246 247 example, the dominant temperature range of (CH<sub>3</sub>)<sub>2</sub>COO + HONO is, 213–275 K for (CH<sub>3</sub>)<sub>2</sub>COO + (H<sub>2</sub>O)<sub>2</sub> and 213–290 K for  $(CH_3)_2COO + H_2O$ . At 20% RH,  $k_{eff}$  of  $(CH_3)_2COO + HONO$  becomes dominant over  $k_{eff}$  of both,  $(CH_3)_2COO + HONO$ 248 249 H<sub>2</sub>O and (CH<sub>3</sub>)<sub>2</sub>COO + (H<sub>2</sub>O)<sub>2</sub> in almost whole temperature range (213–310 K). For example, at 298 K, k<sub>eff</sub> of (CH<sub>3</sub>)<sub>2</sub>COO + HONO is  $\sim 1.8~\text{sec}^{-1}$ , which is 1.6 times and 2.2 times higher than the same for  $(\text{CH}_3)_2\text{COO} + \text{H}_2\text{O}$  and  $(\text{CH}_3)_2\text{COO} + \text{H}_2\text{O}$ 250 251  $(H_2O)_2$ , respectively. This suggests that the major sink of substituted CI can be its reaction with HONO in the atmosphere 252 even in the presence of high humidity and SO<sub>2</sub>. It is important to mention that although substituted CI undergoes a bimolecular 253 reaction with HONO, it is still primarily removed by its unimolecular dissociation, particularly at room temperature. For example, the unimolecular dissociation rate of  $(CH_3)_2COO$  is  $\sim 276 \text{ sec}^{-1}$  at room temperature. Interestingly, the unimolecular rate 254 increases rapidly with temperature, whereas for the bimolecular reaction  $(CH_3)_2COO + HONO$ ,  $k_{eff}$  increases only slightly. 255 As a result, at lower temperatures,  $k_{eff}$  may become comparable to the unimolecular dissociation rate of  $(CH_3)_2COO$ . For ex-256 ample, at 213 K,  $k_{eff}$  and the unimolecular rate constants are 3.80 sec<sup>-1</sup> and 1.82 sec<sup>-1</sup>, respectively. A comparison between 257  $k_{eff}$  and the unimolecular dissociation rate constant of (CH<sub>3</sub>)<sub>2</sub>COO within 213–320 K is provided in Table S6 of the ESI. It 258 259 is evident from Table S6 that under conditions of high HONO concentration and low temperature, the bimolecular reaction of 260 (CH<sub>3</sub>)<sub>2</sub>COO with HONO competes well with its unimolecular dissociation. Finally, it is important to assess the extent to which the title reaction can contribute in resolving the puzzle of mismatch be-261 tween measured and modelled OH<sup>•</sup>/HO<sup>•</sup> concentrations. It is important to mention that during daytime, HONO undergoes 262

rapid photolysis; therefore, its concentration is higher in the absence of light, e.g. at night, indoors, in winter, etc. For example,

263

the photolysis rate of HONO is known to be  $\sim 10^{-3}~{\rm sec}^{-1}$ , which is several orders of magnitude higher than the effective rate 264 constant of its reaction with Criegee intermediates ( $\sim 10^{-7} - 10^{-6} \text{ sec}^{-1}$ , computed using maximum Criegee concentration 265 of  $\sim 10^5$  molecule cm $^{-3}$ ) (Shabin et al., 2023). Therefore, during the peak of daytime, title reaction does not contribute much 266 to OH• production; rather, it can play a key role in nocturnal atmospheric chemistry, specifically at times when both, concen-267 trations of HONO and CI are high, and, at the same time, the presence of light is minimal. To understand the efficiency of the 268 title reaction in affecting OH<sup>•</sup> concentration in a nocturnal environment, we can compare it with NO<sup>•</sup><sub>2</sub> + HO<sup>•</sup><sub>2</sub> reaction, which 269 is a well-known source of OH $^{\bullet}$  at nighttime. The rate constants for CH<sub>2</sub>OO + HONO reaction are  $\sim 2$  times higher compared 270 to  $NO_3^{\bullet} + HO_2^{\bullet}$ . For example, at 298 K, the rate value for  $CH_2OO + HONO$  is  $\sim 7.21 \times 10^{-12}$  cm<sup>3</sup> molecule<sup>-1</sup> sec<sup>-1</sup>, which is 271 almost double compared to the rate value (Rai and Kumar, 2024) for  $NO_3^{\bullet} + HO_2^{\bullet}$ , i.e.,  $\sim 3.36 \times 10^{-12}$  cm<sup>3</sup> molecule<sup>-1</sup> sec<sup>-1</sup>. 272 In the atmosphere, average concentration of both  $NO_3^{\bullet}$  and  $HO_2^{\bullet}$  are  $\sim 10^8$  molecule cm<sup>-3</sup>(Bottorff et al., 2023; Brown and 273 Stutz, 2012), thus combined concentration turns out to be  $\sim 10^{16}$  molecule<sup>2</sup> cm<sup>-6</sup>. Similarly, the combined concentration will 274 be  $\sim 10^{15}$  molecule<sup>2</sup> cm<sup>-6</sup> for CH<sub>2</sub>OO + HONO under high concentrations of CI ( $\sim 10^5$  molecule cm<sup>-3</sup>)(Khan et al., 2018) 275 and HONO ( $\sim 10^{10}$  molecule cm<sup>-3</sup>) (Pawar et al., 2024). It suggests that CH<sub>2</sub>OO + HONO reaction may be somewhat slower 276 in producing OH<sup>•</sup>. However, since the rate of (CH<sub>3</sub>)<sub>2</sub>COO + HONO reaction is one order of magnitude higher compared to 277  $NO_0^{\bullet} + HO_0^{\bullet}$ , we believe both  $NO_0^{\bullet} + HO_0^{\bullet}$  and title reactions should be of similar importance as far as the production of night-278 time OH<sup>•</sup> is concerned. In other words, title reaction has the potential to serve as a significant contributor to OH<sup>•</sup> production 279 in nighttime atmospheric chemistry. 280 Another factor worth noting is, besides OH<sup>•</sup>, the title reaction produces HCHO/(CH<sub>3</sub>)<sub>2</sub>CO, and NO<sup>•</sup><sub>2</sub> as products. It is well 281 282 known that both HCHO/(CH<sub>3</sub>)<sub>2</sub>CO (Gao et al., 2024; Long et al., 2022; Hermans et al., 2004) and NO<sub>2</sub> (Christensen et al., 2004) can act as sinks for HO<sub>2</sub> radicals (corresponding reactions are listed in the box below). It suggests that title reaction has 283

the potential for recycling of  $HO_2^{\bullet} \leftrightarrow OH^{\bullet}$  process. To illustrate the ability of title reaction in recycling  $HO_2^{\bullet} \leftrightarrow OH^{\bullet}$  process,

we have developed a kinetic model consisting of the following reactions (see ESI for the details):

292 This model requires two key components: first, the rate coefficients of the relevant reactions, which have been taken from the recommended literature values (Gao et al., 2024; Hermans et al., 2004; Long et al., 2022; Christensen et al., 2004), and second, a list of realistic initial concentrations of the reactive species involved in  $HO_2^{\bullet} \leftrightarrow OH^{\bullet}$  recycling process (Table S5 of the ESI). We first tracked the change in concentration of  $OH^{\bullet}$  and  $HO_2^{\bullet}$  using the first kinetic model consisting of  $CH_2OO + HONO$  reaction, followed by second model consisting of  $CH_3)_2COO + HONO$  reaction. Initial concentrations of relevant species (HCHO, HONO,  $CH_3)_2CO$ , and  $CH_3)_2CO$ , are chosen based on literature values representing polluted urban conditions

(Vereecken et al., 2012; Pawar et al., 2024). Although the average concentration of OH $^{\bullet}$  can vary within  $\sim 10^4 - 10^6$  molecules 293 cm<sup>-3</sup> in the atmosphere, we have used a modelled value of it in the present work. In CH<sub>2</sub>OO + HONO reaction model, 294 the initial OH $^{\bullet}$  concentration was set to  $\sim 10^4$  molecules cm $^{-3}$ , while in (CH<sub>3</sub>)<sub>2</sub>COO + HONO model, it was set to  $\sim 10^5$ 295 296 molecules cm<sup>-3</sup>. This difference was chosen based on how much OH each reaction is expected to produce when no in situ reactions are taking place from the byproducts of the title reaction. Since (CH<sub>3</sub>)<sub>2</sub>COO + HONO reaction can generate more 297 298 OH, starting with a higher initial concentration helps one observe a noticeable change in OH• levels during the simulation. This makes it easier to observe and compare the effect of OH<sup>•</sup> production between the two reactions. It is important to mention 299 that the maximum concentration of OH $^{\bullet}$  can be taken as  $\sim 10^5$  molecules cm $^{-3}$  in the kinetic model. This is because the 300 production of OH $^{\bullet}$  is limited by the available concentration of CI which can be as high as  $\sim 10^5$  molecules cm $^{-3}$ . Therefore, 301 taking  $OH^{\bullet}$  concentration more than  $\sim 10^5$  molecules cm<sup>-3</sup> would produce no effect on the concentration of  $OH^{\bullet}$ . This also 302 reveals the fact that the title reaction is capable of producing OH• in regions where the concentration of OH• is already low. 303 Similarly, the concentration of NO<sub>2</sub> can vary within  $\sim 10^{10}-10^{12}$  molecules cm<sup>-3</sup> in polluted urban regions. However, in the 304 present model, we have kept it at  $\sim 10^{10}$  molecules cm<sup>-3</sup> in order to observe a clear numerical change in the values of HO<sub>2</sub>. 305 Taking a high concentration of NO<sub>2</sub> ( $\sim 10^{12}$  molecules cm<sup>-3</sup>) would drastically consume HO<sub>2</sub>, and a gradual change would 306 not be observed. 307 We have divided the simulation results into two parts; first we will discuss  $CH_2OO + HONO$  reaction followed by  $(CH_3)_2COO$ 308 + HONO. The model results have been shown in Figure 5. It is evident from Figure 5 that CH<sub>2</sub>OO + HONO reaction increases 309 310 OH• concentration while simultaneously reducing HO<sub>2</sub> concentration. Quantitatively, this reaction increases OH• production by five times its initial value while decreasing HO<sub>2</sub> production by more than one order of magnitude. Furthermore, when 311 we consider dimethyl-substituted Criegee intermediate reaction ((CH<sub>3</sub>)<sub>2</sub>COO + HONO), OH• production has been found to 312 313 increase by only two times compared to its initial concentration, while HO<sup>o</sup><sub>2</sub> production again decreases by the same one order of magnitude (Figure 5). The difference in OH<sup>•</sup> production can be attributed to the fact that, in case of (CH<sub>3</sub>)<sub>2</sub>COO + HONO, 314 the initial  $OH^{\bullet}$  concentration was taken to be  $\sim 10^5$  molecules cm<sup>-3</sup> compared to  $\sim 10^4$  molecules cm<sup>-3</sup> in case of  $CH_2OO$ 315 + HONO. This further strengthens the fact that the effect of title reaction on OH• production will be more pronounced in 316 317 the conditions where OH<sup>•</sup> concentration is lower in the atmosphere, e.g., at night. The overall simulation results suggest that incorporating title reaction into atmospheric models can improve their accuracy in predicting OH• and HO• concentrations. 318 It is important to note that the kinetics model used in the present work is priliminary. However, a more realistic impact of the 319 title reaction on the budget of both OH<sup>•</sup> and HO<sup>•</sup>, requires a more complete modeling. In order to do so, one needs accurate 320 estimation of the rate constants for the reaction of HONO with various important Criegee intermediates. For bigger Criegee 321 322 intermediates, computation will be more costly and require a separate study. In addition, being a HAT reaction, the effect of 323 humidity on the title reaction is also important to build a complete model.

## 324 5 Conclusions

- 325 In this work, we have studied the energetics and kinetics of bimolecular reaction of simple and dimethyl-substituted Criegee 326 with HONO using high-level electronic structure theory and chemical kinetics. Our quantum chemical calculations suggest that both of the reactions are barrierless and kinetic calculations reveal that reaction of substituted Criegee with HONO is  $\sim$ 327 2.6–3.6 times faster than simple Criegee + HONO reaction. By comparing it with other known sinks of CI, we have shown that 328 329 this reaction can serve as a major sink for Criegee intermediates in most of the atmospheric conditions, even in the presence of 330 high humidity and SO<sub>2</sub>. In addition, we have also shown that title reaction can be one of the most important source of OH• in nocturnal atmosphere. In addition, the products of CI + HONO reaction can be a sink for HO<sub>2</sub> radicals, and hence this reaction 331 332 is capable of  $HO_2^{\bullet} \leftrightarrow OH^{\bullet}$  recycling. Consequently, this reaction can be key in fulfilling the gap between the observed OH radicals and modelled values. Although in urban areas, HONO can be the dominant sink of certain CIs. But it is important 333 to notice that larger Criegee intermediates predominantly originate from biogenic volatile organic compounds (BVOCs). On 334 the other hand, HONO concentrations in forested regions are also found to be moderate ( $\sim 10^8$  to  $10^{10}$  molecules cm<sup>-3</sup>). 335 336 Therefore, we believe a separate study is required to understand the fate of larger Criegee intermediates in the presence of HONO. At last, we look forward to the experimental verification of our results. 337
- Author contributions. VJA: Conducted the investigation, Writing—original draft, Formal analysis, curated the data. PKR: Contributed to partial formal analysis, writing, reviewing, and editing the manuscript. PK: Provided supervision, resources, and methodology; conceptualized
- 340 the study; acquired funding; and contributed to the review and editing of the manuscript.
- 341 *Competing interests.* The authors declare that they have no conflict of interest.
- 342 *Acknowledgements.* V.J.A. and P.K.R acknowledge MNIT Jaipur for financial assistance. P.K. acknowledges DST, Govt. of India, for the 343 financial support through the sanctioned project No. EEQ/2023/000351.

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

Figure 1. The potential energy surface for  $CH_2OO + HONO$  reaction (in kcal  $mol^{-1}$ ) obtained at CCSD(T)/CBS//M06-2X/aug-cc-pVTZ level of theory along with optimized geometries of species involved in the reaction.

Figure 2. The potential energy surface for  $(CH_3)_2COO + HONO$  reaction (in kcal mol<sup>-1</sup>) obtained at CCSD(T)/CBS//M06-2X/aug-cc-pVTZ level of theory along with optimized geometries of species involved in the reaction.

**Table 1.** Bimolecular rate constants  $(k_{bi}$ , in cm<sup>3</sup> molecule<sup>-1</sup> sec<sup>-1</sup>) for CH<sub>2</sub>OO/(CH<sub>3</sub>)<sub>2</sub>COO + HONO reaction within the temperature range of 213–320 K.

| T (K) | $\mathbf{k}_{bi}^{CH_2OO}$ | $\mathbf{k}_{bi}^{(CH_3)_2COO}$ |
|-------|----------------------------|---------------------------------|
| 213   | $1.17 \times 10^{-11}$     | $4.28 \times 10^{-11}$          |
| 216   | $1.15 \times 10^{-11}$     | $4.18 \times 10^{-11}$          |
| 219   | $1.13 \times 10^{-11}$     | $4.09 \times 10^{-11}$          |
| 224   | $1.11 \times 10^{-11}$     | $3.94 \times 10^{-11}$          |
| 235   | $1.04 \times 10^{-11}$     | $3.61 \times 10^{-11}$          |
| 250   | $9.58 \times 10^{-12}$     | $3.18 \times 10^{-11}$          |
| 259   | $9.10 \times 10^{-12}$     | $2.94 \times 10^{-11}$          |
| 265   | $8.79 \times 10^{-12}$     | $2.78 \times 10^{-11}$          |
| 278   | $8.14 \times 10^{-12}$     | $2.47 \times 10^{-11}$          |
| 280   | $8.04 \times 10^{-12}$     | $2.42 \times 10^{-11}$          |
| 290   | $7.57 \times 10^{-12}$     | $2.20 \times 10^{-11}$          |
| 298   | $7.21 \times 10^{-12}$     | $2.03 \times 10^{-11}$          |
| 300   | $7.13 \times 10^{-12}$     | $1.99 \times 10^{-11}$          |
| 310   | $6.70 \times 10^{-12}$     | $1.80 \times 10^{-11}$          |
| 320   | $6.30 \times 10^{-12}$     | $1.63 \times 10^{-11}$          |

Figure 3. Effective rate constant comparison ( $k_{eff}$ , in sec<sup>-1</sup>) of CH<sub>2</sub>OO + HONO with the  $k_{eff}$  of previously known sinks of CH<sub>2</sub>OO. a. Values are taken from reference (Lin et al., 2016)

b. Values are taken from reference (Onel et al., 2021)

**Figure 4.** Effective rate constant comparison  $(k_{eff}$ , in  $\sec^{-1})$  of  $(CH_3)_2COO + HONO$  with the  $k_{eff}$  of previously known sinks of  $(CH_3)_2COO$ .

- a. Values are taken from reference (Vereecken et al., 2017)
- b. Values are taken from reference (Smith et al., 2016)

Figure 5. Top panel: Concentration profiles of  $HO_2^{\bullet}$  and  $OH^{\bullet}$  using  $CH_2OO + HONO$  reaction into the model. Bottom panel: Concentration profiles of  $HO_2^{\bullet}$  and  $OH^{\bullet}$  using  $(CH_3)_2COO + HONO$  reaction into the model.