# Peer review of "Criegee + HONO reaction: the dominant sink of Criegee, and the missing non-photolytic source of OH•"

_EGUsphere, 2025_

## Referee Comment (RC1)

Manuscript ID egusphere-2025-1364, entitled "Criegee + HONO reaction: the dominant sink of Criegee, and the missing non-photolytic source of OH". The work is interesting to understand the atmospheric oxidation capacity of Criegee in the atmosphere. Although the investigation claimed a new source of OH and the sink of $HO_2$ and suggested that HONO may be the dominant sink of Criegee, the claims could not be based on scientific foundation. Therefore, the importance of Criegee + HONO should be turned down.

(1) The atmospheric lifetimes of $CH_2OO$ with water dimer had been detailedly investigations from theoretical and experimental methods. Please read the reference (J. Am. Chem. Soc. 2021, 143, 8402-8413). The atmospheric lifetime is $2.12 \times 10^{-4}$ s at 0 km in Table 7 in J. Am. Chem. Soc. 2021, 143, 8402-8413. Although the atmospheric lifetime of $CH_2OO$ with water dimer is very long in the stratosphere, the concentrations of Criegee intermediates are very low at altitude above 15 km. Therefore, the importance of Criegee intermediates occurs in the troposphere. According to Table 1 in the present work, I assume that the concentration of HONO is about $10^{10}$ molecules cm$^{-3}$ in the troposphere, which leads to the atmospheric lifetime of $CH_2OO$ with HONO is about $10^2$ s. This shows that HONO does not make any contribution to the sink of $CH_2OO$. In addition, Criegee intermediates are mainly produced from the ozonolysis of BVOCs, while HONO is mainly produced at urban regions.

(2) The second issue is computational methods. In fact, there are dozens of papers that have shown that post-CCSD(T) calculations are required to obtain quantitative barrier heights for the reactions including Criegee intermediates. Although I have to admit the introduction of the calculations will extremely increase the computational costs, it should be clearly explained and reviewed in the present progress. This is very helpful for potential readers to know the progress. Please read these articles (J. Am. Chem. Soc.2025, 147 (14), 12263-12272.; Atmos. Environ. 2025, 341, 120928.; Research **2024**, 7, 0525.; Fundam. Res. **2024**, 4 (5), 1216-1224.; Proc. Natl. Acad. Sci. USA 2018, 115, 6135-6140. And so on).

(3) In kinetics calculations, there are still lots of factors that do not consider such as recrossing effects, torsional anharmonicity, and anharmonicity. In addition, the low energy barrier, what is the rate-determining step. I guess that the formed pre-reactive complex is the rate-determining step like Criegee reaction with HCOOH, Therefore, VRC-TST is necessary for the barrierless process.

(4) Lines "Our study also suggests that HONO has the potential to become the most dominant sink of Criegee intermediate, surpassing SO2 and water dimer, even in high humid condition", it is not validated.

(5) Lines "the bimolecular reaction paths can be the main sink of sCI (Osborn and Taatjes, 2015; Lin et al., 2015; Sheps et al., 2014; Vereecken and Francisco, 2012)." Some important key references have been missed such as . J. Am. Chem. Soc. 2016, 138, 14409-14422. and J. Am. Chem. Soc. 2021, 143, 8402-8413.

(6) Kinetics methods should be moved into computational section.

(7) It needs to add some tables for showing the atmospheric lifetimes of Criegee intermediates with $H_2O$, $(H_2O)_2$, $SO_2$, HCOOH, and HONO as the function of altitude.

---

## Author Comment (AC1)

**Response to the Reviewer's Comments**

August 1, 2025

**1 REFEREE REPORT**

**Comment: The atmospheric lifetimes of $CH_2OO$ with water dimer had been detailedly investigations from theoretical and experimental methods. Please read the reference (J. Am. Chem. Soc. 2021, 143, 8402-8413). The atmospheric lifetime is $2.12\times10^{-4}$ sec at 0 km in Table 7 in J. Am. Chem. Soc. 2021, 143, 8402-8413. Although the atmospheric lifetime of $CH_2OO$ with water dimer is very long in the stratosphere, the concentrations of Criegee intermediates are very low at altitude above 15 km. Therefore, the importance of Criegee intermediates occurs in the troposphere. According to Table 1 in the present work, I assume that the concentration of HONO is about $10^{10}$ molecules $cm^{-3}$ in the troposphere, which leads to the atmospheric lifetime of $CH_2OO$ with HONO is about $10^2$ sec. This shows that HONO does not make any contribution to the sink of $CH_2OO$. In addition, Criegee intermediates are mainly produced from the ozonolysis of BVOCs, while HONO is mainly produced at urban regions.**

**Reply**: We agree with the reviewer that under high humid condition, HONO is not a major sink for simple $CH_2OO$ in the atmosphere; rather, it plays a dominant role in the removal of substituted Criegee intermediates, i.e., $(CH_3)_2COO$. This is clearly illustrated in Figure 3 and 4 of the main manuscript. It is evident from Figure 3 that at 100% relative humidity (RH), the reaction of $CH_2OO$ with $(H_2O)_2$ is dominant across the entire temperature range studied (213–320 K). But it is important to mention that the concentration of $(H_2O)_2$ as well as $H_2O$ greatly depends on the relative humidity (RH) and temperature. For example, at 20% RH, the effective rate constant $(k_{eff})$ for $CH_2OO$ + HONO becomes comparable to the same for $CH_2OO$ + $(H_2O)_2$ and $CH_2OO$ + $H_2O$ reactions in the lower temperature ranges of 213–235 K and 213–260 K, respectively. This suggests that although $CH_2OO$ + HONO is a minor sink under typical tropospheric conditions, it can become relevant under specific atmospheric conditions. We

have discussed it in the revised manuscript on page 7.

For the second concern of the referee regarding the sources of Criegee and HONO, it is worth mentioning that simpler Criegee intermediates (those Criegee which has less than four carbon atoms) are produced from the both sources, i.e. ozonolysis of biogenic volatile organic compounds (BVOCs) in forested environments and from anthropogenic sources in urban areas. For example, $CH_2OO$ (formed from 23 VOCs) has 20% production from anthropogenic sources and 19% from biogenic sources. Similarly, $(CH_3)_2COO$ (formed from 10 VOCs) has 28% anthropogenic production and 9% biogenic production. Thus, the two Criegee intermediates selected in the present work are representative of species emitted from both biogenic and anthropogenic sources. We have added a short discussion of it in the conclusion of the revised manuscript. As far as HONO is concerned, although HONO is primarily generated in urban regions, several field measurements have reported a reasonable HONO concentrations ($\sim 10^8$ to $10^{10}$ molecules $cm^{-3}$) even in forested areas[6, 1, 10, 18, 5, 15, 11, 12, 19].

**Comment: The second issue is computational methods. In fact, there are dozens of papers that have shown that post-CCSD(T) calculations are required to obtain quantitative barrier heights for the reactions including Criegee intermediates. Although I have to admit the introduction of the calculations will extremely increase the computational costs, it should be clearly explained and reviewed in the present progress. This is very helpful for potential readers to know the progress. Please read these articles (J. Am. Chem. Soc.2025, 147 (14), 12263-12272.; Atmos. Environ. 2025, 341, 120928.;Research 2024, 7, 0525.; Fundam. Res. 2024, 4 (5), 1216-1224.; Proc. Natl. Acad. Sci. USA 2018, 115, 6135-6140. And so on).**

**Reply**: As indicated by the reviewer, post-CCSD(T) calculations are indeed computationally very demanding. Still, to assess the uncertainty in the energetics due to the exclusion of post-CCSD(T) corrections, we have carried out CCSDT(Q)/CBS calculations for the smaller Criegee intermediate reaction ($CH_2OO$ + HONO). We have focused on key stationary points, i.e., the reactant complex (RC) and transition state (TS). The different components of post-CCSD(T) corrections ($\delta_T$ and $\delta_T(Q)$) are provided in Table S7 of the ESI. It is evident from Table S7 that post-CCSD(T) corrections have made only minor changes in the calculated energetics of the $CH_2OO$ + HONO reaction. In fact, post-CCSD(T) corrections have reduced the stabilization energy of RC by only $\sim 0.54$ kcal mol$^{-1}$; on the other hand, they have raised the barrier height by a similar amount, i.e., 0.67 kcal mol$^{-1}$, which lies within the chemical accuracy. This suggests that our CCSD(T)/CBS//M06-2X/aug-cc-pVTZ level of theory is both reliable and

computationally efficient for studying the title reaction. To further confirm this, we have also computed the rate constants using the post-CCSD(T) level energetics and found negligible changes in the rate constant of $CH_2OO$ + HONO reaction. For example, at 298 K, the rate constant decreased slightly from $\sim 7.2 \times 10^{-12}$ $cm^3$ molecule$^{-1}$ sec$^{-1}$ to $\sim 5.5 \times 10^{-12}$ $cm^3$ molecule$^{-1}$ sec$^{-1}$. This further supports the reliability of our computational approach. We have discussed it in the revised manuscript on page 6 and line 180.

**Comment: In kinetics calculations, there are still lots of factors that do not consider such as recrossing effects, torsional anharmonicity, and anharmonicity. In addition, the low energy barrier, what is the rate-determining step. I guess that the formed pre-reactive complex is the rate-determining step like Criegee reaction with HCOOH, Therefore, VRC-TST is necessary for the barrier-less process.**

**Reply**: The reviewer is right that in the title reaction, the formation of the pre-reactive complex is the rate-determining step. Since this step is barrierless, a variational treatment is essential for obtaining accurate rate coefficients. To account for this, we have employed KTOOLS code as implemented in MultiWell suite of programs, which uses variational transition state theory (VTST) for the barrierless association process. The inputs for the KTOOLS are potential energy surface scans along the coordinate describing the dissociation of RC to isolated reactants. Therefore, a variational approach is explicitly incorporated in our kinetic calculations for the initial step. We have now added few lines in the manuscript to make it more clearer. In fact, this methodology is consistent with previous studies on bimolecular reactions of Criegee intermediates, where similar kinetic treatments have been successfully applied [17, 8, 4, 3, 13, 7, 9, 14, 16, 2]. In addition, in the revised manuscript, we have also included a deterministic eigenvalue-eigenvector-based approach, specifically the Bartis-Widom method (implemented in MESMER program) to estimate rate constants. In addition, to account for torsional anharmonicity, we have also performed a relaxed potential energy scan of the torsional rotation along N–O bond of HONO moiety. The resulting torsional potential has been used to model hindered internal rotation (HIR). This correction led to negligible changes in the calculated rate constants. We have added the details of it on page 4.

**Comment: Lines "Our study also suggests that HONO has the potential to become the most dominant sink of Criegee intermediate, surpassing $SO_2$ and water dimer, even in high humid condition", it is not validated.**

**Reply**: We agree with the reviewer that this statement is not valid for

all types of CI + HONO reaction. In fact, we made this statement for our substituted $(CH_3)_2COO$ + HONO reaction. In Figure 4 of the manuscript, it can be clearly seen that the $k_{eff}$ of $(CH_3)_2COO$ + HONO reaction is dominant over almost the entire temperature range, even in the presence of $(H_2O)_2$ at RH 100% and $SO_2$. Therefore, this statement is quite valid for the dimethyl-substituted Criegee intermediate but not for all the Criegee intermediate. Now in the revised manuscript, we have corrected that statement in the abstract.

**Comment: Lines "the bimolecular reaction paths can be the main sink of sCI (Osborn and Taatjes, 2015; Lin et al., 2015; Sheps et al., 2014; Vereecken and Francisco, 2012)." Some important key references have been missed such as . J. Am. Chem. Soc. 2016, 138, 14409-14422. and J. Am. Chem. Soc. 2021, 143, 8402-8413.**

**Reply**: Thanks for the references. These references was indeed helpful. As per the reviewer's advice, we have duly cited these references in the revised manuscript.

**Comment: Kinetics methods should be moved into computational section.**

**Reply**: As per the reviewer's advice, we have now moved the kinetics methods to the methodology section of the revised manuscript.

**Comment: It needs to add some tables for showing the atmospheric lifetimes of Criegee intermediates with $H_2O$, $(H_2O)_2$, $SO_2$, HCOOH, and HONO as the function of altitude.**

**Reply**: Unfortunately, we could not find any literature containing relevant data on HONO concentrations as a function of altitude (perhaps due to difficulty in the field measurements). Therefore, we have avoided estimating altitude-dependent lifetimes of Criegee intermediates with HONO.

**References**

[1] Karin Acker, Detlev Möller, Wolfgang Wieprecht, Franz X Meixner, Birger Bohn, Stefan Gilge, Christian Plass-Dülmer, and Harald Berresheim. Strong daytime production of oh from hno$_2$ at a rural mountain site. *Geophys. Res. Lett.*, 33(2), 2006.

[2] Vishva Jeet Anand and Pradeep Kumar. Mechanistic insight into the $n_2o + o(^1d, ^3p)$ reaction: role of post-ccsd(t) corrections and non-adiabatic effects. *Phys. Chem. Chem. Phys.*, 25(48):33119–33129, 2023.

[3] Rabi Chhantyal-Pun, Robin J Shannon, David P Tew, Rebecca L Caravan, Marta Duchi, Callum Wong, Aidan Ingham, Charlotte Feldman, Max R McGillen, M Anwar H Khan, et al. Experimental and computational studies of criegee intermediate reactions with $nh_3$ and $ch_3nh_2$. *Phys. Chem. Chem. Phys.*, 21(26):14042–14052, 2019.

[4] Amit Debnath and Balla Rajakumar. Bimolecular kinetics of criegee intermediate ($ch_2oo$) with 2-pentanone: experimental and theoretical analysis in atmospheric conditions. *Environmental Science and Pollution Research*, pages 1–10, 2024.

[5] Yi He, Xianliang Zhou, Jian Hou, Honglian Gao, and Steven B Bertman. Importance of dew in controlling the air-surface exchange of hono in rural forested environments. *Geophys. Res. Lett.*, 33(2), 2006.

[6] Saewung Kim, S-Y Kim, Meehye Lee, Heeyoun Shim, GM Wolfe, Alex B Guenther, Amy He, Youdeog Hong, and Jinseok Han. Impact of isoprene and hono chemistry on ozone and ovoc formation in a semirural south korean forest. *Atmos. Chem. Phys.*, 15(8):4357–4371, 2015.

[7] Stephen J Klippenstein and James A Miller. From the time-dependent, multiple-well master equation to phenomenological rate coefficients. *J. Phys. Chem. A.*, 106(40):9267–9277, 2002.

[8] Keith T Kuwata, Emily J Guinn, Matthew R Hermes, Jenna A Fernandez, Jon M Mathison, and Ke Huang. A computational re-examination of the criegee intermediate–sulfur dioxide reaction. *J. Phys. Chem. A.*, 119(41):10316–10335, 2015.

[9] Jari Peltola, Prasenjit Seal, Anni Inkilä, and Arkke Eskola. Time-resolved, broadband uv-absorption spectrometry measurements of criegee intermediate kinetics using a new photolytic precursor: unimolecular decomposition of $ch_2oo$ and its reaction with formic acid. *Phys. Chem. Chem. Phys.*, 22(21):11797–11808, 2020.

[10] X Ren, H Gao, X Zhou, JD Crounse, PO Wennberg, EC Browne, BW LaFranchi, RC Cohen, M McKay, AH Goldstein, et al. Measurement of atmospheric nitrous acid at bodgett forest during bearpex2007. *Atmos. Chem. Phys.*, 10(13):6283–6294, 2010.

[11] Xinrong Ren, William H Brune, Angelique Oliger, Andrew R Metcalf, James B Simpas, Terry Shirley, James J Schwab, Chunhong Bai, Utpal

Roychowdhury, Yongquan Li, et al. Oh, ho$_2$, and oh reactivity during the pmtacs–ny whiteface mountain 2002 campaign: Observations and model comparison. *J. Geophys. Res. Atmos.*, 111(D10), 2006.

[12] Alberto Rondon and Eugenio Sanhueza. High hono atmospheric concentrations during vegetation burning in the tropical savannah. *Tellus B*, 41(4):474–477, 1989.

[13] Saptarshi Sarkar and Biman Bandyopadhyay. Singlet ($^1\delta_g$) o$_2$ as an efficient tropospheric oxidizing agent: the gas phase reaction with the simplest criegee intermediate. *Phys. Chem. Chem. Phys.*, 22(35):19870–19876, 2020.

[14] Daniel Stone, Kendrew Au, Samantha Sime, Diogo J Medeiros, Mark Blitz, Paul W Seakins, Zachary Decker, and Leonid Sheps. Unimolecular decomposition kinetics of the stabilised criegee intermediates ch$_2$oo and cd$_2$oo. *Phys. Chem. Chem. Phys.*, 20(38):24940–24954, 2018.

[15] Hang Su, Ya Fang Cheng, Min Shao, Dong Feng Gao, Zhong Ying Yu, Li Min Zeng, Jacob Slanina, Yuan Hang Zhang, and Alfred Wiedensohler. Nitrous acid (hono) and its daytime sources at a rural site during the 2004 pride-prd experiment in china. *J. Geophys. Res. Atmos.*, 113(D14), 2008.

[16] Michael F Vansco, Rebecca L Caravan, Kristen Zuraski, Frank AF Winiberg, Kendrew Au, Nisalak Trongsiriwat, Patrick J Walsh, David L Osborn, Carl J Percival, M Anwar H Khan, et al. Experimental evidence of dioxole unimolecular decay pathway for isoprene-derived criegee intermediates. *J. Phys. Chem. A.*, 124(18):3542–3554, 2020.

[17] Nathan AI Watson and Joseph M Beames. Bimolecular sinks of criegee intermediates derived from hydrofluoroolefins–a computational analysis. *Environ. Sci. Atmos*, 3(10):1460–1484, 2023.

[18] N Zhang, X Zhou, S Bertman, D Tang, M Alaghmand, PB Shepson, and MA Carroll. Measurements of ambient hono concentrations and vertical hono flux above a northern michigan forest canopy. *Atmos. Chem. Phys.*, 12(17):8285–8296, 2012.

[19] Xianliang Zhou, Ning Zhang, Michaela TerAvest, David Tang, Jian Hou, Steve Bertman, Marjan Alaghmand, Paul B Shepson, Mary Anne Carroll, Stephen Griffith, et al. Nitric acid photolysis on forest canopy surface as a source for tropospheric nitrous acid. *Nat. Geosci.*, 4(7):440–443, 2011.

---

## Author Comment (AC2)

**Response to the Reviewer's Comments**

August 1, 2025

**1 REFEREE REPORT**

**Comment: The description of their rate calculations seems to imply that they evaluate the overall rate constant as a product of a separately calculated rate for the formation of RC1 and a master equation calculation for the branching in the thermal dissociation of RC1 (between forward reaction, kuni, and back dissociation to reactants). This product of terms would be appropriate if RC1 was being formed in the high pressure limit. But it most certainly is not. In this case, the master equation should instead be used to directly obtain the rate constant for proceeding from the reactants to the bimolecular products.**

   **Reply**: As per the reviewer's suggestion, we have now employed a traditional master equation approach using MESMER software package to estimate the rate constants. In the revised manuscript, we have provided MESMER-based calculations, where pseudo-first-order rate constants are determined directly from the reactants to the isolated products using the master equation method. The rate constants obtained from MESMER are provided in Table 1 of the revised manuscript.

**Comment: There is also some possibility that the PCi complexes are collisionally stabilized since they are fairly deep wells on their PESs. Such stabilization would be important as it would reduce the rate of forming OH. Thus, properly formulated master equations should include the PCi complexes and some rate for their decomposition.**

   **Reply**: In the revised manuscript, we have now added the reaction path from the product complex (PC) to the isolated products in the rate constant calculations using MESMER software package. The estimated effective forward rate values after incorporating PC $\rightarrow$ isolated products step comes out to be almost same compared to the previous one (without adding this step).

**Comment: The authors refer to a KTOOLS code for estimating the formation rate. The authors should also briefly describe the physical assumption behind the calculation in KTOOLS. This point is significant because their formation rates appear to be about an order of magnitude less than what would be expected.**

Reply: We agree with the reviewer that special care is needed in estimating the RC formation rate constant, as this is the rate-determining step. Since this association step is barrierless, a variational treatment is essential for obtaining accurate rate coefficients. To account for this, we have employed KTOOLS code as implemented in the MultiWell suite of programs, which uses variational transition state theory (VTST) for the barrierless reaction. The inputs for KTOOLS are potential energy surface scans along the coordinate describing the dissociation of RC to isolated reactants. Each point on the potential energy surface serves as a trial transition state; KTOOLS searches for the transition state for which the reaction flux is minimized. To address the referee's concern, in the revised manuscript, we have refined the potential energy surface scan for this step. The reviewer is correct that the RC formation rate now becomes one order of magnitude higher compared to the previous work. We have now added a brief discussion of this on page 4 of the revised manuscript.

**Comment: The wells and TSs (and perhaps the reactants) appear to have hindered rotational modes, some of which might have multiple distinct minima. At the very least, the authors should describe how they treated those torsional motions, and whether or not they searched for multiple torsional minima to ensure they had found the global minimum conformational states.**

Reply: We agree that the reactant complex (RC) and transition state (TS) have hindered rotational motions, and there may be multiple conformations due to different torsional angles. To take this into account, we have used HinderedRotorQM1D model in the MESMER software to compute the rate constants. Specifically, we performed a one-dimensional potential energy scan of OH torsion along the N–O bond in both the RC and TS. The scan covered the full 0° to 360° range. The resulting energy profile was used to calculate the hindered rotor partition functions. During this scan, we found local minima in both the RC and TS, which suggests that our originally optimized structures correspond to the global minimum conformers. After incorporating this hindered rotor correction, the computed rate constants are found to be almost same compared to rigid-rotor harmonic oscillator (RRHO) treatment, which indicates that torsional anharmonicity has minimal impact on the overall kinetics for this system. We have added

the details of it on page 4 line 121.

**Comment: It appears that the TS energy reported here for $CH_2OO$ + HONO is about 8 kcal/mol below what was reported in an earlier report from the same group. This is rather odd since the electronic structure methods are very similar. Should I presume that the uncertainty in the energy is truly that large. Some comment on this discrepancy is needed, and ideally the authors would provide some indication of the expected uncertainty in their energies.**

**Reply**: The previous study investigated a different path of the same reaction (that lead to HPMN product). Therefore, the TS of previous study is different from that found in the present work. As a result, the calculated barrier height in the present work is differing ($\sim 8$ kcal mol$^{-1}$ lower in energy). However, to further check for uncertainty in the energetics, we have carried out post-CCSD(T) calculations (CCSDT(Q)/CBS) for the smaller Criegee intermediate reaction i.e., $CH_2OO$ + HONO reaction, focusing on key stationary points, i.e., RC and TS. The obtained post-CCSD(T) corrections have made only minor changes in the calculated energetics of $CH_2OO$ + HONO reaction. In fact, post-CCSD(T) corrections have reduced the stabilization energy of RC by only $\sim 0.54$ kcal mol$^{-1}$; on the other hand, they have raised the barrier height by a similar amount, i.e., 0.67 kcal mol$^{-1}$, which lies well within the range of chemical accuracy.

**Comment: It is well known that $CH_2OO$ has significant multireference character that often disappears in the TSs for its reaction. This commonly results in about a 1 kcal/mol raising of the barrier heights relative to CCSD(T)/CBS estimates. Some comment on this shortcoming in their estimates would be helpful.**

**Reply**: We have dealt with this problem in two ways. First, we have employed a well-established specialized method for multireference systems, i.e., incorporating post-CCSD(T) corrections to validate the energetics. Second, we have performed an uncertainty analysis by taking $\pm 1$ kcal mol$^{-1}$ uncertainty in the reaction barriers as well as well depths (detailed discussion can be found on page 6 and line 175 of the revised manuscript). Both of these approaches introduce only slight changes in the rate constants, which suggests that energy uncertainty due to the multireference character of Criegee intermediates is not going to alter the overall conclusions of the present work.

**Comment: The authors claim that 0.04 angstrom geometry errors clearly suggest that M062X geometries are accurate. For this statement to be true, the authors should provide some estimate of how large an error could arise from such bond length errors.**

In principle, that is straightforward from some consideration of typical force constants. Simply stating that the geometry errors are small is not helpful. Similarly, the authors claim that 250 cm-1 frequency errors imply that M062X is appropriate for frequency calculations. From my experience, those sorts of frequency errors are extraordinarily large, and would make me wonder if I had done something wrong. My expectation is that they could yield order of magnitude sorts of errors in the predicted rates. Some more appropriate discussion of the meaning of those shortcomings is needed.

**Reply**: To address the reviewer's concern, we have calculated the uncertainties associated with the computed rate constant due to an error of $250$ cm$^{-1}$ ($\sim 0.7$ kcal mol$^{-1}$) in the energetics of the reaction. We have assumed this much uncertainty ($\pm$ 1 kcal mol$^{-1}$) in well depths as well as in reaction barriers and estimated uncertainty in the rate constants at 298 K for simple Criegee + HONO reaction. Due to $\pm$ 1 kcal mol$^{-1}$ uncertainty in the reaction barriers and well depths, the maximum deviation in the rate constant is $\sim 7.21^{+4.67}_{-3.65} \times 10^{-12}$ cm$^3$ molecule$^{-1}$ sec$^{-1}$ ($\pm 1$ reaction barriers) and $\sim 7.21^{+0.45}_{-0.45} \times 10^{-12}$ cm$^3$ molecule$^{-1}$ sec$^{-1}$ ($\pm 1$ well depths), respectively. It suggests that this much of uncertainty is not going to affect the overall conclusion of the present work. We have discussed this point on page 6 and line 187 of the revised manuscript.

**Comment: The focus on just the bimolecular rate constants in their discussion of the effective rate constants is misleading. For the $(CH_3)_2COO$ case, the unimolecular decomposition rate near room temperature is about 400 sec$^{-1}$, which swamps their effective bimolecular rates. With that in mind, their suggestion that the CI + HONO reactions are the major sink for the CI requires some indication as to how rapidly the unimolecular decay rates decrease with temperature.**

**Reply**: The reviewer is right that the unimolecular rate of $(CH_3)_2COO$ is higher at room temperature, i.e., $\sim 276$ sec$^{-1}$. But it is important to mention that the unimolecular rate increases rapidly with temperature, whereas for the bimolecular reaction (CH3)$_2$COO + HONO, k$_{eff}$ increases only slightly. As a result, at lower temperatures, k$_{eff}$ becomes comparable to the unimolecular dissociation rate of (CH3)$_2$COO. For example, at 213 K, k$_{eff}$ and unimolecular rate constants are 3.80 sec$^{-1}$ and 1.82 sec$^{-1}$, respectively. A comparision between k$_{eff}$ and unimolecular dissociation rate constant of $(CH_3)_2COO$ within 213–320 K is provided in Table S6 of the ESI. It is evident from Table S6 that under conditions of high HONO concentration and low temperature, the bimolecular reaction of $(CH_3)_2COO$ with HONO competes well with its unimolecular dissociation. We have included this discussion in the revised manuscript on page 8 and line 251.

**Comment: It would be helpful to have some estimate of the expected uncertainty in their rate predictions.**

**Reply**: As per the referee's suggestion, we have added a discussion on the uncertainties associated with the computed rate constant on page 6 of the main manuscript.

**Comment: The actual HONO concentration used in their keff calculations should be explicitly stated.**

**Reply**: In the revised manuscript, we have stated explicitly that the concentration of HONO is $8.9 \times 10^{10}$ molecule $cm^{-3}$.

**Comment: The model simulations are limited enough in scope that I consider them to be highly speculative at best.**

**Reply**: We understand the reviewer's concern about the limited scope of our model simulations. Our primary aim was to provide qualitative insight and mechanistic understanding under a defined set of assumptions and conditions. A more comprehensive study including all Criegee intermediates (CIs), unimolecular, and bimolecular sinks would indeed be necessary for a broader analysis. Such an investigation, however, would require a separate and detailed study. We have included this discussion also in revised manuscript on page 10 and line 319.

---

## Author Response (AR2)

**Response to the Reviewer's Comments**

October 7, 2025

**1 REFEREE REPORT**

Comment: As pointed out by the first reviewer the present work only suggests that the Criegee + HONO reaction is the dominant sink of Criegee for  $(CH_3)_2COO$  and not at all for  $CH_2OO$ . And even for  $(CH_3)_2COO$  it is only dominant at best at temperatures below 225 K. For  $CH_2OO$ , Fig. 3 shows that the  $CH_2OO + SO_2$  reaction is always a larger sink than the title reaction. For  $(CH_3)_2COO$ , the unimolecular loss is the dominant sink except perhaps below 225 K. The authors need to make a much more serious effort to remove all of their overstatements. For example, the title needs to be revised, the concluding statements need to be further revised, and the plot in Fig. 4 must include the unimolecular loss rate. The authors must reread the whole manuscript to make sure that any such overstatements have been removed. It is important for a paper to have a correct physical interpretation and tone in addition to correct data.

**Reply**: We agree with the reviewer that HONO acts as a sink for  $(CH_3)_2COO$ , but not for  $CH_2OO$ . We further acknowledge that HONO is dominant sink of  $(CH_3)_2COO$  at temperatures below 225 K. In the revised manuscript, we have updated Figure 4 to include the unimolecular rate of  $(CH_3)_2COO$ . In the revised manuscript, we also make sure that from title to conclusion our language represent the real picture rather any overstatement.

Comment: The authors need to either improve their discussion of why they believe the M06-2X/aug-cc-pVTZ is an effective approach for determining the stationary point geometries and vibrational frequencies, or simply remove all of that discussion. An error in the geometry of 0.04 Å converts to a 2 kcal/mol error for a typical force constant of 2000 N/m. Is that what the authors deem an appropriate error for an accurate geometry? Similarly, an error of 250 cm-1 in the frequency correlates with anywhere from a

few % error to a factor of two or more error in the partition function (depending on which frequency is in error by that amount). Is a factor of two error in the rate prediction an indication that the method is adequate for kinetics calculations? And that is for just one vibrational mode. What is the cumulative effect of such errors for many modes? Oddly, I actually believe that the M06-2X/aug-cc-pVTZ is a suitable enough method for predicting the kinetics for this reaction, but the authors arguments are so poorly presented that one can readily come to the opposite conclusion. A correct statement would be that, prior literature studies suggest 2 sigma uncertainties of  $\sim 2~\rm kcal/mol$  in CCSD(T)/CBS//M06-2X barrier heights and factor of two sorts of uncertainties in partition function ratios. It would be much better to simply say something like that.

**Reply**: We thank the reviewer for the suggestion. In the revised manuscript, we simply state that the CCSD(T)/CBS//M06-2X/aug-cc-pVTZ level of theory, was chosen based on previous studies where this level of theory was found to be reasonable in similar reaction.

Comment: The description of the VTST calculations with KTOOLS is missing key details. Ordinarily a VTST calculation requires some description of both the potential energies along some form of a minimum energy path and the vibrational frequencies for the orthogonal motions along the reaction path. The latter are not described at all. Furthermore, the authors provide no indication of the electronic structure method that was used to map the PES. Nor do they describe the method used to scan the PES – is it some sort of distinguished reaction coordinate analysis?

Reply: In the revised manuscript, we have provided the details of the VTST calculations. The minimum energy path (MEP) was obtained by manually scanning along the reaction coordinate, with geometries optimized at the M06-2X/aug-cc-pVTZ level of theory using Gaussian 16. Single point energies at each point were improved at the CCSD(T)/CBS level. The vibrational frequencies orthogonal to the reaction coordinate were computed at each point along the MEP and employed in the VTST using the KTOOLS package. These frequencies are provided in Table S9 of the ESI.

Comment: The ILT for the reaction from the PC to the products also needs some description of the presumed temperature dependence.

**Reply**: In the revised manuscript, now we have mentioned that for ILT,

the rate constants are assumed to be independent of temperature. (on page 4 on line 115)

Comment: It's nice to know that the torsional scans did not find lower global minimal conformers. Nevertheless, some description should still be provided regarding how extensive your search for conformers was initially. Was some manual search of the conformational space performed? Did you just accept whatever conformer was first found?

Reply: We performed both, an automated scan and a manual search, for lower energy conformers using the Gaussian 16 package. All conformers were fully optimized at the M06-2X/aug-cc-pVTZ level of theory, and their single point energies were computed at the CCSD(T)/CBS level. Among these, no conformers were found with energies lower than that of the reported conformer. Now we have mention it in the revised manuscript. Therefore, we believe that the reported conformer represents the global minimum.

Comment: Why would the L-J parameters for a CH2OO-HONO complex correspond to the average of those for CH2OO and HONO. Shouldn't they correspond more closely to some sort of sum of the parameters. Certainly the size of the complex is not the average of the individual sizes of the components. This treatment should be revised and the master equation calculations repeated.

**Reply**: We agree with the reviewer. In the revised manuscript, we have fitted the Lennard-Jones (L-J) parameters. These parameters are used to describe the collisional interactions with the bath gas. To obtain them, we performed a bond distance scan between the reactive complex (RC) and the bath gas molecule, and subsequently fitted the interaction energy with the bond distance in the Lennard-Jones potential. The fitted plot provided the corresponding L-J parameter values.

Comment: On. p. 5 – why do you describe the reaction as occurring in two steps? Isn't it three steps – formation of RC1, passage through TS1 to form PC1, then decomposition of PC1 to products. This proper physical description as three steps is important because there may be some stabilization of PC1. On a related note, the authors should mention whether or not there is there any stabilization of either reactant or product complexes? With 45 kcal/mol exothermicity and 27 kcal/mol bond dissociation energies there is particularly some possibility of the product complexes being stabilized.

**Reply**: In the revised manuscript, we have mentioned that the reaction is a three step process. The product complex is  $\sim 44.70 \text{ kcal mol}^{-1}$  stable with respect to the isolated reactants, while the bond dissociation energy required to convert the PC into the final isolated product is  $\sim 27.4 \text{ kcal mol}^{-1}$ . In addition we have computed Gibbs free energy profile of the reaction at 298 K, (Figure S2 of ESI). The Gibbs free energy profile suggests that the isolated products are  $\sim 2.5 \text{ kcal mol}^{-1}$  lower in free energy relative to the PC.

Comment: On line 146 RC should be RC1. On Line 158 RC should be RC2.

**Reply**: In the revised manuscript, we have corrected "RC" to "RC1" on line 146 and to "RC2" on line 158.

Comment: The capture rates reported in Table S3 are still smaller than I would have naively guessed by about a factor of 4. This is likely an artifact of some limitation in the VTST calculation, but since those are not described in appropriate detail it is hard to judge what the issue is.

Reply: We agree with the reviewer that the capture rate may be faster than our calculated value. The capture rate is rate determining step and essentially at collision limit. The VTST calculated rate is sensitive to the the minimum energy path scan points. So this could be artifact of VTST. We have mention the details of VTST (energy and frequency) on Table S9 of ESI.

Comment: A proper citation (i.e., likely to one of his websites) should be provided to the actual source of the Ruscic et al. thermochemical values for the reaction energy.

**Reply**: In the revised manuscript, we have added the appropriate citation to the official Active Thermochemical Tables (ATcT).

Comment:It appears that the rate calculations were performed without the CCSDT(Q) correction. Why? Do the authors somehow believe that including that correction would not improve the accuracy of the predictions. If so, they should state that.

**Reply**: Due to the limited computational facilities, we were able to perform CCSDT(Q)/CBS calculation only for the simplest Criegee not for the dimethyl substituted Criegee. In the revised manuscript, for simplest Criegee, we have compare the rate constant estimated at CCSDT(Q)/CBS PES with CCSD(T)/CBS values to estimate the uncertainty in rate constant

due the multi-reference nature of reaction. Our post-CCSD(T) corrections are below 1 kcal  $\text{mol}^{-1}$  and the estimated rate constants are also similar. For example, at 298 K, the bimolecular rate constants calculated at the post-CCSD(T) and CCSD(T)/CBS levels are  $5.53 \times 10^{-12}$  and  $7.21 \times 10^{-12}$  cm3 molecule-1 sec-1, respectively.

Comment: The discussion of the uncertainties on p. 6 lacks any discussion of the uncertainties arising from inadequate partition function evaluations. I would estimate that such uncertainties are at least a factor of two and probably more like a factor of 4. Indeed, as I mention above, I expect that the true capture rate would probably be a factor of four larger, and almost certainly a factor of two larger. It appears that their uncertainty analysis does not consider the uncertainties in this capture rate. Furthermore, I strongly suspect that the barrier height 2 sigma uncertainties are more like 2 kcal/mol. Also, since the more relevant part of the prediction is for low temperature, they should indicate how the uncertainties change with decreasing temperature.

**Reply**: We agree with the reviewer. In the revised manuscript, we have included  $2\sigma$  ( $\pm$  2 kcal mol-1) uncertainties in the error analysis of the rate constants. Accordingly, we have also added a discussion of the uncertainties arising from the partition function evaluations. This discussion has been incorporated into the revised manuscript (on page 6 on line 192).

Comment: The paragraph on p. 7 starting on line 219 first misleads the reader to think that the  $CH_2OO + HONO$  is dominant at combined low temperature and low RH and then corrects that to indicate that  $CH_2OO + SO_2$  is always dominant over  $CH_2OO + HONO$ . Just start off with the proper statement that  $CH_2OO + HONO$  is never the dominant sink, or even major sink (it looks like the maximum is about 20% at about 240 K and low RH). Then the details can be discussed. That is the proper primary finding of your calculations and the more clearly it is stated the better.

**Reply**: In the revised manuscript, we have rectified our statement and revised the manuscript accordingly. "In Figure 3, we have compared the  $k_{eff}$  of  $CH_2OO + HONO$  with the  $k_{eff}$  of  $CH_2OO + H_2O/(H_2O)_2/SO_2$  reactions. Figure 3 shows, HONO is not a major sink of  $CH_2OO$ . It is evident from Figure 3 that at 100% RH,  $k_{eff}$  of  $CH_2OO + (H_2O)_2$  is the dominant reaction across the entire temperature range (213–320 K). As far

as  $CH_2OO + SO_2$  reaction is concerned, its  $k_{eff}$  values are  $\sim 5$  times higher than that of  $CH_2OO + HONO$  reaction within the whole temperature range. At 20% RH,  $k_{eff}$  for  $CH_2OO + (H_2O)_2$  and  $CH_2OO + H_2O$  remain dominant at higher temperatures, specifically within 235–320 K and 260–320 K, respectively. However, at lower temperatures,  $k_{eff}$  of  $CH_2OO + HONO$  becomes dominant, surpassing both,  $CH_2OO + (H_2O)_2$  and  $CH_2OO + H_2O$  in the range of 213–235 K and 213–260 K, respectively. It is indicating that  $CH_2OO + HONO$  reaction is a minor contributor compared to the other sinks of Criegee intermediates. "

Comment: Table S6 is central to the authors argument that the  $HONO + (CH_3)_2COO$  is the dominant sink at low temperature. Their data for the unimolecular dissociation of  $(CH_3)_2COO$  in that Table needs some citation as to the source of the data. That data also needs to be directly plotted in Fig. 4. Then the discussion of Fig. 4 on p. 8 should again start with the primary conclusion that only below 225 K is the  $(CH_3)_2COO + HONO$  reaction predicted to be the dominant sink of  $(CH_3)_2COO$ . Then more details can be provided.

**Reply**: We thank the reviewer for the suggestions. In the revised manuscript, we have cited the relevant Table and modified Figure 4 to include the unimolecular rate. We also corrected our statement on page 8 as follows: "It is evident from Figure 4 that at temperatures below 225 K, HONO is the dominant sink for (CH3)2 COO. At temperatures above 225 K, unimolecular dissociation becomes the major sink. As far as bimolecular sink of  $(CH_3)_2COO$  is concerned HONO can be a major sink for  $(CH_3)_2COO$ . It is evident from Figure 4 that at 100% RH,  $k_{eff}$  of  $(CH_3)_2COO + HONO$ can dominate over  $k_{eff}$  of  $(CH_3)_2COO + H_2O$  and  $(CH_3)_2COO + (H_2O)_2$ for a relatively wider range of temperatures. For example, the dominant temperature range of  $(CH_3)_2COO + HONO$  is, 213–275 K for  $(CH_3)_2COO$  $+ (H_2O)_2$  and 213-290 K for  $(CH_3)_2COO + H_2O$ . At 20% RH,  $k_{eff}$  of  $(CH_3)_2COO + HONO$  becomes dominant over  $k_{eff}$  of both,  $(CH_3)_2COO +$  $H_2O$  and  $(CH_3)_2COO + (H_2O)_2$  in almost whole temperature range (213– 310 K). For example, at 298 K,  $k_{eff}$  of  $(CH_3)_2COO + HONO$  is  $\sim 1.8$  sec-1, which is 1.6 times and 2.2 times higher than the same for  $(CH_3)_2COO +$  $H_2O$  and  $(CH_3)_2COO + (H_2O)_2$ , respectively "

Comment: Again, the statement in the conclusion "By comparing it with other known sinks of CI, we have shown that this reaction can serve as a major sink for Criegee intermediates in most of the atmospheric conditions" is not in fact true.

**Reply**: We correct our statement in the revised manuscript. "By comparing it with other known sinks of CI, we have shown that HONO can serve as a major bimolecular sink for bigger Criegee intermediates ((CH3)2COO) and minor contributor at low humidity and low temperature for simple CH2OO."

**Response to the Reviewer's Comments**

October 7, 2025

**1 REFEREE REPORT**

Comment: Unfortunately, the relevant articles are still missed such as J. Am. Chem. Soc. 2016, 138, 14409-14422. and J. Am. Chem. Soc. 2021, 143, 8402-8413.

**Reply**: We thank the reviewer's careful assessment. We have now cited them appropriately in the revised manuscript.

---

## Author Response (AR4)

**Response to the Reviewer's Comments**

October 15, 2025

**1 REFEREE REPORT**

**Comment: Although it is not clearly stated in the manuscript, it seems that the authors don't seem to see any stabilization of the RCs or the PCs. In that case, the values used for the LJ parameters are not particularly relevant to the conclusions of this paper. Nevertheless, to avoid future confusion for other authors, I note that the LJ parameters reported here are strange, and the description of how they are calculated seems incomplete. The words suggest that they directly calculate the LJ parameters for the RC...$N_2$ interaction. But then, how can the size, sigma (which presumably represents the distance between the center-of-mass of the RC and $N_2$ at the minimum of the complex) be only 2.6 Å, when the separation between the CI and the HONO in the RC itself is on the order of 3 Å? This makes no sense. Also, if they are directly calculating the LJ parameters for the RC...$N_2$ interaction then why do they report values for the bath gas alone. Those values shouldn't even enter into the kinetic analysis.**

    **Reply:** The reviewer is correct that we did not find any stabilization of the RCs or the PCs. We thank the reviewer for pointing out the discrepancy in the LJ parameters, which were due to a high error in the fitting parameters. In the revised manuscript, we have refined the fitting to reduce the error. We now obtain reasonable LJ parameters within an acceptable level of MUD. The newly fitted $\epsilon$ and $\sigma$ values are 895.5 K and 3.1 Å, respectively. The reviewer is also correct that there is no need to report the LJ parameters of the bath gas; therefore, in the revised manuscript, we have removed that sentence from the methodology section.

**Comment: The authors should report what value they presumed or calculated (I still can't tell if it was calculated or presumed-it seems like it must be the latter since no data is provided for its MEP) for the capture rate for the process from products to the**

**PC complexes. Or do they not include this process in their master equation and simply presume the PC complexes dissociate?**

    **Reply**: We would like to clarify that the capture rate for the process from the products to the PC complexes was obtained using the ILT method. We have now provided the corresponding rate in Table S10 of the ESI. The fitted Arrhenius parameters derived from these rate values were used in the master equation calculations.

**Comment: At line 269 the authors state that "keff increases only slightly". I think this should say "keff decreases only slightly" since the context of the sentence is for increasing temperature.**

    **Reply**: We have now rectified that sentence in the revised manuscript.

**Comment: The left-hand y-axes in Figure 5 need some improvement.**

    **Reply**: As per refere's advice, we have refined the left-hand y-axes of Figure 5.